# Immunization of Children under 2 Years Old in the Coastal Hadhramaut Governorate, Yemen, during Public Health Emergencies: A Trend Analysis of 2013–2020

**DOI:** 10.3390/vaccines12030311

**Published:** 2024-03-15

**Authors:** Suha Ali Batarfi, Rosnah Sutan, Halim Ismail, Abdulla Salem Bin-Ghouth

**Affiliations:** 1Department of Public Health Medicine, Medical Faculty, Universiti Kebangsaan Malaysia, Kuala Lumpur 56000, Malaysia; 2Community Medicine Department, Hadhramout University College of Medicine and Health Sciences, Al Mukalla 10587, Yemen

**Keywords:** childhood immunization, pandemic, crisis, preparedness plan, Yemen

## Abstract

Although immunization is one of the most successful and cost-effective interventions that prevents millions of infant and child deaths yearly, it has failed to achieve its intended goals in some low-income countries. Yemen is currently experiencing the most extreme humanitarian crisis globally, which has affected health and worsened its economy and political governance instability. There are few reports on Yemeni vaccination statuses. The present study aimed to investigate the effect of the public health emergency crises on childhood immunization in Yemen. A retrospective descriptive study was conducted in the Coastal Hadhramaut Governorate, Yemen. Secondary data from governorate annual reports for 2013–2020 were extracted. The assessment of the annual immunization coverage rate according to each vaccine was tabulated. The analysis revealed that the 2013–2019 vaccination coverage in Coastal Hadhramaut demonstrated an increasing trend. However, vaccination coverage decreased for all vaccines in 2015–2016 and 2020. Although all three doses of the pentavalent vaccine demonstrated >85% coverage in all years, the coverage of the first and second doses decreased in 2016, and the coverage of all doses decreased in 2020 during the COVID-19 pandemic. Public health emergencies negatively affected routine immunization coverage in Yemen. The trend correlated with the humanitarian crisis and other research findings in Yemen. The national response to public health threats during emergency crises must involve strengthening the program for monitoring and evaluating vaccine-preventable diseases.

## 1. Introduction

Primary prevention through vaccination programs is one of the most effective and cost-efficient means of preventing infant and child deaths. However, limited access and funding for vaccine programs affect its implementation, as was seen in some underserved populations where vaccine programs failed to reach the intended goals [1]. Each year, vaccination averts an estimated 2–3 million child deaths from vaccine-preventable diseases (VPDs) [2]. Vaccination is vital to attain Sustainable Development Goal (SDG) objectives, specifically SDG 3 (Good Health and Well-being). It has a broader reach than any other healthcare or social service, forming the cornerstone of primary healthcare systems and serving as a significant method for achieving universal health coverage [3].

Yemen is a low-income Arab nation with an unstable healthcare system and poor socioeconomic conditions. Yemen comprises 22 governorates over 333 districts. The largest governorate in Yemen is Hadhramaut, which is affected by the internal displacement of residents from other governorates [4]. The Expanded Program of Immunization (EPI) in Yemen was initiated in 1979 to reduce morbidity and mortality associated with VPDs. The EPI targets children under 1 year old, pregnant women, and women of reproductive age [5]. Childhood vaccination coverage in Yemen improved significantly since the EPI began. However, the most recent Yemen National Health and Demographic Survey in 2013 reported that only 42.6% of children aged 12–23 months were fully vaccinated, while 16% did not receive any vaccination [6].

Nevertheless, coverage gaps persist between regions, countries, and within countries. The World Health Organization (WHO) estimated that approximately 17% of infants worldwide (22.7 million infants) did not receive the diphtheria, tetanus, and pertussis (DTP) vaccine in 2020 [1,7]. Routine vaccination coverage in several WHO Eastern Mediterranean Region (EMRO) countries markedly improved during the past two decades. However, vaccination declined in some countries (Yemen, Syria, and Iraq) due to prevailing geopolitical circumstances, which decreased regional DTP dose 3 (DTP3) coverage from 85% (2010) to 80% (2016) [8]. Furthermore, Southeast Asia and the EMRO recorded the largest decline in DTP3 coverage in 2020 [8]. The United Nations Children’s Fund (UNICEF) and WHO reported that the vaccination coverage for Yemen in 2018 was 65% based on DTP3, which remained below the 90% that every country should have achieved by 2020 [9].

Disasters, both natural and human-made, are increasing worldwide and have serious consequences for humanity and economies [10]. Children and adolescents are the most vulnerable groups affected by the consequences of disasters [11]. Thus, all countries should establish and maintain a preparedness plan to overcome such emergencies. Although many health interventions and research priorities have been published for reference, studies in the Middle East indicated that this topic requires much work [12]. Yemen has been exposed to many natural and man-made disasters in the past decade, such as earthquakes, cyclones, flooding, internal conflict, civil war, and pandemics [13]. It has experienced conflict since 2011, when anti-government protests began. In March 2015, the ongoing Yemen civil war began when Houthi rebels, a Shi’a group, seized control of the Yemeni capital, Sana’a [4,14].

Both the conflict and civil war negatively affected the national infrastructure and resulted in further political and social instability. Millions of people were displaced from their homes and often faced challenges accessing healthcare services, including basic child immunization [15]. More than 50% of Yemeni health facilities are currently unable to provide healthcare services following the crisis [16]. The risk of violence and fear of attacks prevented families from accessing healthcare, including immunization services. Disruptions to the vaccine supply chain, damage to cold-storage facilities, and limited access to remote areas also contributed to the decreased vaccination coverage in Yemen [17]. Thus, the constant conflict in Yemen has caused a widespread humanitarian catastrophe and destroyed an already fragile health system [18].

The COVID-19 pandemic had broad effects and affected most aspects of human life. The pandemic affected health system management because of staffing deployment to control the pandemic, and disrupted the delivery of most health services following movement restriction orders, including immunization [19]. Furthermore, the pandemic affected routine vaccination globally, where the coverage of DTP3 and the polio vaccine decreased from 86% in 2019 to 83% in 2020. This decrease led to the largest number of unvaccinated children since 2009 (22.7 million), an increase of 3.7 million from 2019. Additionally, the global coverage of the first dose of measles-containing vaccine (MCV1) decreased from 86% in 2019 to 84% in 2020 [7]. Several studies addressed the effect of the pandemic on child health globally, especially childhood vaccination, and confirmed the decline in routine vaccination coverage in children in numerous countries, especially in low- and middle-income countries (LMICs) [20].

Yemen was weakened due to the ongoing civil war, and the COVID-19 pandemic exacerbated the situation [21,22]. Yemen faces a complex multifaceted situation; the war and the pandemic have significantly affected the achievement of SDG 3 and other SDGs, such as SDG 1 (No Poverty), SDG 2 (Zero Hunger), and SDG 6 (Clean Water and Sanitation). Figure 1 illustrates the main conflict events and health issues in 2011–2020 that influenced the deterioration of health facilities and services.

Studying routine immunization trends can reveal vital insights and inform and aid EPI stakeholders in developing plans to minimize the number of unvaccinated children who dropped out of the EPI during the Yemen crisis [23]. Only one study, by Torbosh et al. [14], was conducted to assess the one-year impact of war on childhood immunization in Yemen. To the best of our knowledge, no other published studies address the effect of a public health crisis (e.g., continuous war and pandemic) in Yemen. Therefore, this study intends to address the issue of child immunization coverage by analyzing the child immunization coverage trend in Hadhramaut Governorate, Yemen, during the emergency crisis (especially the war and the COVID-19 pandemic) in 2013–2020.

## 2. Materials and Methods

### 2.1. Study Design and Setting

This paper is part (phase 1) of 3 phases of a mixed-method study exploring parental vaccine hesitancy and childhood vaccination in coastal Hadhramaut, Yemen. The protocol of the methodology has already been published [5]. For the phase 1 study, a retrospective descriptive study was performed with written permission to use secondary data obtained from the Coastal Hadhramaut Health and Population Affairs Office in 2021, after being granted ethic approval on 18th February 2021. The study focused on Hadhramaut Governorate as it is the largest governorate in Yemen and the one most affected by war-displaced people. The Hadhramaut Governorate administration is divided into the valley/desert (Wadi Hadhramaut) and coast (Coastal Hadhramaut) (Figure 2) [24]. This study focused on Coastal Hadhramaut, which consists of 12 districts: Al-Mukalla, Al-Dees, Al-Shahr, Al-dulaia, Broome-Mayfa’a, Doan, Gail Bawazeer, Ghail bin Yumin, Hajer, Ridh and Gosaiar, rural Al-Mukalla, and Yeabeth. Coastal Hadhramaut overlooks the Arabian Sea to the south and has approximately 120 km of coastline. An estimated 994,771 people lived in Coastal Hadhramaut in 2021, and the yearly population growth rate is 3.02% [25].

### 2.2. Vaccination Process in Yemen

Following EPI initiation in 1979, routine immunization services in Yemen are offered for free through fixed vaccination posts in health facilities (health centers or units) and an outreach vaccination strategy depending on the resident’s proximity to the facility. Initially, the EPI aimed to reduce child mortality and morbidity for 6 VPDs and later expanded to cover 11 VPDs (tuberculosis, poliomyelitis, diphtheria, pertussis, tetanus, hepatitis B, pneumococcal infections, meningitis, rotavirus diarrheal diseases, measles, and rubella). The WHO recommended that the EPI strategy is to achieve no less than 90% vaccination coverage nationally and at least 80% at the governorate level. In the Coastal Hadhramaut Governorate, 166 governmental health facilities provide immunization services, which are insufficient to meet the population’s basic needs, especially with the influx of people displaced from nearby districts [26]. The high fertility rate in Yemen contributes to high service demands with a limited healthcare staffing ratio [25].

Vaccination documentation is routinely charted daily at the health facilities using tally sheet forms and routine vaccination registration records. The data from each health facility are compiled monthly and forwarded to the district immunization manager, who collates and summarizes them as a monthly district immunization report. Subsequently, the 12 districts deliver their monthly district immunization reports to the Department of EPI in the Health and Population Affairs Office—Coastal Hadhramaut. Later, the data are entered into an Excel sheet for analysis and grouped to compile the data for the governorate. Then, the compilation is delivered to the Department of EPI at the Ministry of Public Health and Population to be reported as an annual report [5,26].

### 2.3. Data Collection and Analysis

The present study collected secondary data via soft-copy annual reports and available data in Excel format from 2013 to 2020. The researcher rechecked the data by examining a hard copy of each district immunization report to ensure that the data were valid. The researcher ensured that there were no missing data and no data entry errors, and that the district report data matched the Excel data. A trend analysis was conducted by assessing the yearly vaccination coverage for each vaccine type by district (for multidose vaccines, the coverage rate for each dose was estimated). The WHO defines vaccination coverage as “the proportion of a given population that has been vaccinated in a given period” [27]. The vaccination coverage rate (mostly expressed as a percentage) was calculated by dividing the total number of children who received a specific vaccine during the reporting year (the numerator) by the target population during that year (the denominator) and multiplying the result by 100 [26]. The EPI data use the population projection for the current years based on 2004 census data as there are no available census data since then. The internal displacement of people from the neighboring Governorates during the war has caused the vaccination coverage to be slightly more than 100%. The factors contributing to coverage of more than 100% were due to reporting errors that resulted from the population estimate being lower than the actual population’s current size and temporary population increases due to the presence of internal migration. Capping vaccination coverage at 100% is carried out for reporting in the present study to prevent misunderstanding on data interpretation with a coverage of more than 100%. Table 1 presents the routine immunization schedule, as presented in the analysis: one dose of bacillus Calmette-Guérin (BCG); doses 1–3 of the oral polio vaccine (OPV), the pneumococcal conjugate vaccine (PCV), and a pentavalent vaccine that protects against diphtheria, tetanus, pertussis, hepatitis B, and *Hemophilus influenzae* type B (Hib); doses 1 and 2 of the measles-rubella combination vaccine (MR); doses 1 and 2 of the rotavirus vaccine; and one dose of the inactivated poliovirus vaccine (IPV).

### 2.4. Ethical Consideration

Ethical approval was obtained, and the Coastal Hadhramaut Health and Population Affairs Office general manager granted written permission to use the EPI department data. The Research Ethics Committee of the Faculty of Medicine, Universiti Kebangsaan Malaysia, granted ethical approval (project code: FF-2021-051).

## 3. Results

Anonymous data obtained from the EPI department were recorded according to vaccination type and year. Data from 2013 to 2020 in Excel format were obtained together with the hard copy of each district immunization report. The researcher verified the data for completeness and validity. There were no data entry errors, no missing data, and the same data for each district were available in both hard- and soft-copy formats. The annual reports presented the vaccination coverage for each Coastal Hadhramaut district.

Generally, Coastal Hadhramaut vaccination coverage increased for most vaccines from 2013 to 2019 (Figure 3 and Figure 4). However, vaccination coverage decreased for all vaccines in 2020. Furthermore, BCG, MR, and rotavirus vaccine coverage decreased in 2015–2016 in addition to the decrease in 2020.

### 3.1. BCG Vaccine Coverage

The BCG vaccine is administered immediately after birth and is the first vaccine type administered in Yemen. Most children receive the BCG vaccine at birth. At the age of 6 weeks, children receive OPV dose 1, the pentavalent and pneumococcal vaccines. BCG vaccine coverage in Coastal Hadhramaut decreased slightly in 2014, with a marked decrease in 2015 (−11%) (Figure 3a). Subsequently, BCG vaccine coverage began to increase from 2016 to 2019. In 2020, BCG coverage decreased (−15%) compared to 2019. The district-level analysis revealed a similar increasing trend in BCG vaccine coverage until 2020 when the coverage began to decrease. Decreased BCG coverage occurred in most districts in 2015 (Appendix A). The Al-Mukalla district demonstrated persistently high BCG vaccine coverage for all years (>80%), while rural Al-Mukalla, Doan, Al-Dulaia, Hajer, and Yeabeth demonstrated persistently low coverage for all years (<80%). The other districts demonstrated varied BCG vaccine coverage, where most had good coverage after 2015.

### 3.2. IPV Coverage

IPV was introduced to the routine immunization program in Yemen at the end of 2015 and is administered to children in one dose at the age of 14 weeks [6]. As with other vaccines, IPV coverage increased in Coastal Hadhramaut until 2020, before it sharply decreased (−14%) (Figure 3b). District-level analysis revealed that all districts demonstrated decreased coverage in 2020. The Al-Mukalla, Ridh and Gosaiar, Ghail Bawazeer, and Ghail bin Yumin districts exhibited persistently good coverage, while the Al-Dees, Doan, Al-Dulaia, and Yeabeth districts exhibited constantly low IPV coverage (Appendix A).

### 3.3. MR Vaccine Coverage

Yemeni children receive MR vaccine dose 1 (MR1) at 9 months of age, and dose 2 at 18 months. In 2014, the coverage of both doses 1 and 2 of the MR vaccine in Coastal Hadhramaut decreased (−5%) as compared to 2013 (Figure 4a). Another decline was recorded in 2016 and 2017 for both doses. Subsequently, vaccine coverage increased over time until 2019, when the coverage of both doses began to decrease in 2020 (Figure 4a).

District-level analysis revealed that despite the increased MR1 coverage over the years, most districts exhibited low vaccine coverage in 2020, and only four districts had coverage > 80% (Al-Mukalla, Ghail Bawazeer, Ghail bin Yumin, and Yeabeth). Al-Dulaia and Hajer demonstrated persistently low coverage of MR1 for all years, while other districts demonstrated varied coverage, with low coverage in some years and good coverage in later years. However, MR2 coverage was persistently low in all districts over time (Appendix A).

### 3.4. Rotavirus Vaccine Coverage

Yemeni children receive two doses of the rotavirus vaccine. Doses 1 and 2 are administered when the child is 6 and 10 weeks old, respectively. Although the coverage of both rotavirus vaccine doses increased over time in Coastal Hadhramaut, coverage in 2016 decreased by 5% compared to 2015 and decreased by 9% in 2020 compared to 2019 (Figure 4b). The district-level analysis revealed that the coverage of both doses of the rotavirus vaccine increased over time. Nevertheless, most districts had <80% coverage between 2013 and 2016, except Al-Mukalla district, which had persistently good coverage (>80%) in all years (2013–2020). After 2016, all districts demonstrated increased vaccine coverage, where most had >80% coverage (Appendix A).

### 3.5. OPV, Pentavalent, and Pneumococcal Vaccine Coverage

The Yemen EPI schedule states that doses 1, 2, and 3 of the OPV, pentavalent, and pneumococcal vaccines are administered when the child is 6, 10, and 14 weeks old, respectively. These vaccines were administered on the same occasion, and we determined that the data for each vaccine dose were similar. Accordingly, the data are presented as one vaccine type (pentavalent) over three doses (Figure 4c). Although the coverage of all three doses in all years in Coastal Hadhramaut was >85%, the coverage of doses 1 and 2 decreased in 2016. Subsequently, their coverage increased together with that of the other vaccines until 2020, when all doses decreased by approximately 9% compared to 2019.

The district-level analysis demonstrated that all districts exhibited increased coverage of pentavalent vaccine doses 1 and 2 from 2013 to 2019. Furthermore, all districts had high coverage except Hajer in 2013–2017, Broome-Mayfa’a in 2013 and 2014, Al-Shahr in 2016, Ghail Bawazeer in 2015 and 2016, Yeabeth in 2017, and Al-Dees in 2013 and 2014, which had <80% coverage. In 2020, all districts demonstrated decreased coverage compared to previous years. Generally, increased coverage was recorded for pentavalent vaccine dose 3 over the years in all districts, except for Al-Dees in 2014, Hajer in 2014 and 2015, and Yeabeth in 2017, which demonstrated decreased coverage. In 2020, all districts demonstrated decreased coverage compared to 2019, where Al-Shahr and Al-Dees had <80% coverage while all other districts had a high coverage rate (Appendix A).

## 4. Discussion

Our results and evidence in the literature indicate a possible connection between the effect of war and the COVID-19 pandemic on the achievement of immunization status. Vaccination coverage is a significant indicator for tracking and guiding immunization programs at all levels. In the present study, we described the vaccination coverage trend in children living in Coastal Hadhramaut, Yemen, in 2013–2020 (before and after the civil war and 1 year during the COVID-19 pandemic). Our results revealed decreased coverage for most vaccines in Coastal Hadhramaut in 2015 and 2016. In 2020, vaccination coverage was decreased for all vaccine types. The reasons for the decrease are multifactorial and were exacerbated by the ongoing conflict and pandemic. Nevertheless, the results can be attributed to several factors, such as parents, healthcare system accessibility, vaccine stock availability, and the public health emergencies that disrupted health program management and monitoring.

### 4.1. Public Health Emergency and Vaccination Coverage in Yemen

#### 4.1.1. Conflict and Humanitarian Crisis

The Yemen civil war began in 2015–2016. While the Hadhramaut Governorate was not directly involved in the war, the political, economic, and social war affected all Yemeni regions [4]. Furthermore, the local Al-Qaida branch seized Al-Mukalla, the Hadhramaut Governorate capital, and the surrounding regions in 2015, and controlled them for approximately 1 year. Consequently, the situation isolated Coastal Hadhramaut from direct governmental control [22].

During a conflict, human requirements are restricted to basic life needs. Together with the unstable local conditions, immunization was considered an additional requirement that could be delayed or even abandoned. Furthermore, the thousands of people who had escaped war and migrated to Hadhramaut increased general local society demands and complicated the living situation. Moreover, the difficulty in tracking and reaching displaced populations contributed to the decreased vaccine coverage. Yemeni healthcare infrastructure was weak even before the conflict, where access to healthcare facilities was limited, especially in rural areas. The conflict exacerbated the situation and led to the destruction or closure of many health centers and clinics [18]. Although several immunization campaigns were held in 2014–2020, the lack of functioning healthcare facilities and trained personnel prevented the provision of many health services, including immunization [17]. Additionally, a lack of awareness of immunization importance, incorrect ideas, misinformation, and rumors regarding vaccination, especially in communities with internal conflict, affect parents’ decisions about vaccinating their children, as parents are concerned about vaccine safety and efficacy specifically during the conflict, leading to a reluctance to vaccinate their children [14,28].

Our results agreed with the results of other studies conducted in Yemen, where most concluded that vaccination coverage decreased markedly after the war began [4,14,29,30]. Torbosh et al. conducted a study using EPI administrative data for all governorates in 2012–2015 (before and 1 year after the civil war) to determine the effect of the 2015 war on the immunization coverage of children under 1 year old in Yemen. The analysis of the 2012–2015 vaccination coverage data revealed that the war negatively affected immunization coverage. The analysis determined that the pentavalent-3 vaccine coverage was 82% in 2012 and 88% in 2014, the measles vaccine coverage was 70% in 2012 and 75% in 2014, while the 2015 coverage for the pentavalent-3 and measles vaccines was 84% and 66%, respectively. That study concluded that there were greater reductions in governorates with armed confrontations, while governorates that did not have armed confrontations had increased coverage [14]. Other studies reported that immunization coverage significantly affected vaccination services, which mainly explained the current increase in cases of VPDs, such as measles, diphtheria, and diarrheal diseases throughout the country, which led to outbreaks [4,16,31]. In 2017, a diphtheria outbreak was confirmed in Yemen, where 2203 probable diphtheria cases (including 116 deaths) were reported [16]. Furthermore, measles outbreaks were reported in different years in Yemen, with more than 3000 suspected cases being reported throughout the country in 2018 [32]. UNICEF launched a nationwide vaccination campaign against childhood diseases, targeting 5.4 million Yemeni children. However, the vaccination coverage reported remained unsatisfactory, although the Hadhramaut Governorate was one of the governorates with better and more stable social conditions at that time [33].

#### 4.1.2. COVID-19 Pandemic

Following the detection of the first Yemeni case of COVID-19 in Hadhramaut in April 2020, the imposed curfew, fear of infection, and general vaccine hesitancy all reduced public interest in any other health care than protecting themselves against COVID-19 infection. These conditions persisted in 2020 and 2021 until the COVID-19 vaccine became available in the country, which brought hope and relief. The literature also supported the reports of decreased immunization coverage during the COVID-19 pandemic, where the pandemic affected routine immunization globally and a greater effect was expected in LMICs [23,34,35,36,37]. A recent study conducted in Ecuador (an LMIC) that aimed to assess the vaccination coverage during the pandemic in 2020 reported a decreased number of vaccine doses and vaccination coverage for infants for the rotavirus, poliovirus, PCV, and pentavalent vaccines as compared to previous years [20]. The study also explained that the causes of decreased vaccination coverage could be related to parental choices to use another method for immunity, misconception, national lockdowns, mobilization restrictions, and fear of infection [20]. Similarly, a study conducted in Pakistan to investigate the effect of the COVID-19 lockdown on routine immunization reported the same conclusion [23]. Based on WHO, UNICEF, Global Alliance for Vaccine Initiative (GAVI), and Sabin Vaccine Institute data, COVID-19 hindering vaccination services in at least 68 countries placed approximately 80 million children under the age of 1 year at risk of childhood VPDs such as diphtheria, pertussis, polio, hepatitis B, pneumococcus, measles, and rotavirus infections [19]. Furthermore, lockdowns, social distancing, the fear of contagion, and supply chain interruption compelled parents to delay their children’s routine immunization [23,38].

Our results indicated that immunization coverage increased in 2016–2019, which was attributed to the improved social situation in Hadhramaut, as Houthi rebels were expelled from most of the southern governorates, and Al-Qaida was ousted in 2016 [22]. Thus, life returned to an acceptable state although the war continued. The people had become accustomed to the current conditions, and a large proportion of displaced people had returned to their original regions. These changes provided residents with the opportunity to consider their children’s health and seek immunization centers to resume or start their children’s immunization programs. The healthcare authorities cooperated with the WHO and UNICEF to conduct several vaccination campaigns, especially in 2018–2019 [39].

### 4.2. The Effect of Civil War and the COVID-19 Pandemic on SDG 3 Achievement

Both the civil war and the COVID-19 pandemic greatly affected the achievement of SDG 3 in Yemen (and hence Hadhramaut). The Yemeni healthcare system collapsed during the ongoing conflict, and many healthcare centers were destroyed or became non-functional. The damaged infrastructure, road closures, and ongoing violence restricted access to healthcare services, which limited the availability of essential healthcare services and affected progress toward SDG 3 [18]. The exacerbated humanitarian crisis led to widespread food insecurity, malnutrition, and displacement, which weakened the immune systems of many Yemenis and rendered them more susceptible to disease. Thus, the weakened healthcare system, limited access to clean water and sanitation, malnutrition, and overcrowded living conditions created favorable conditions for disease spread. Consequently, many communicable disease outbreaks occurred in 2014–2020, such as cholera, measles, dengue fever, diphtheria, and COVID-19 [16,22,30]. These outbreaks presented additional challenges to achieving SDG 3 [4]. Furthermore, immunization programs in Yemen were disrupted, which decreased immunization coverage [14], affected Yemeni children’s well-being, and hindered progress toward SDG 3 and the achievement of other interconnected SDGs [3].

Although this study presented a significant overview of childhood vaccination coverage during the main crisis in Yemen in the past decade, it was also subject to some limitations. First, we used data from EPI departments and obtained complete data for only 8 years (2013–2020). The 2 years of data pre-2013 (2011–2012) were incomplete and thus they were excluded from the analysis. Secondly, as in other LMICs, no census has been conducted recently in Yemen; the last census was conducted in 2004. The EPI data use the population projection for the current years based on 2004 census data as no census data have been available since then. We used the EPI data in the same manner in the present study and might have overestimated or underestimated the vaccination coverage due to the estimation of the target population.

## 5. Conclusions

The humanitarian conflict and COVID-19 pandemic negatively affected childhood vaccination coverage in Coastal Hadhramaut, Yemen. The trend correlated with the stage of the humanitarian crisis and previous studies conducted in Yemen. Efforts to address the challenges require comprehensive approaches (humanitarian assistance, peacebuilding initiatives, and long-term development strategies) to restore and strengthen the healthcare system and achieve sustainable progress towards SDG 3. Public health professionals should address misconceptions and concerns by increasing the awareness of the importance of vaccinations and providing clear, culturally sensitive information on vaccine benefits and safety. The continuous and timely assessment of vaccination coverage is required to respond to the emergency crises in Yemen. Furthermore, countries, especially LMICs such as Yemen, should establish and maintain a preparedness plan to overcome emergencies. Moreover, further research to evaluate the consequences of the public health crisis on childhood vaccination coverage is recommended.

## Figures and Tables

**Figure 1 vaccines-12-00311-f001:**
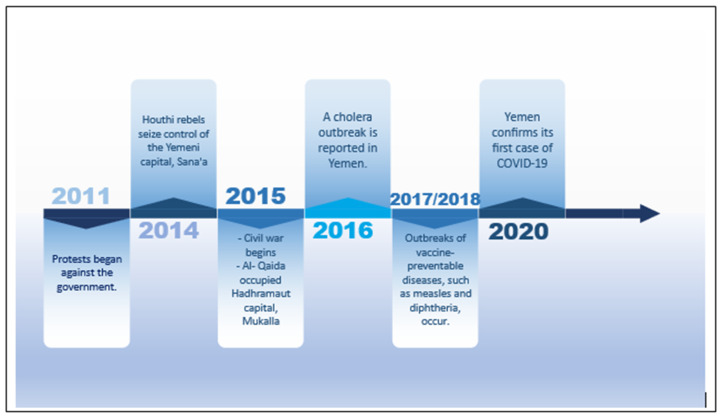
Milestones of main conflict events and their relationship with health issues in Yemen in 2011–2020.

**Figure 2 vaccines-12-00311-f002:**
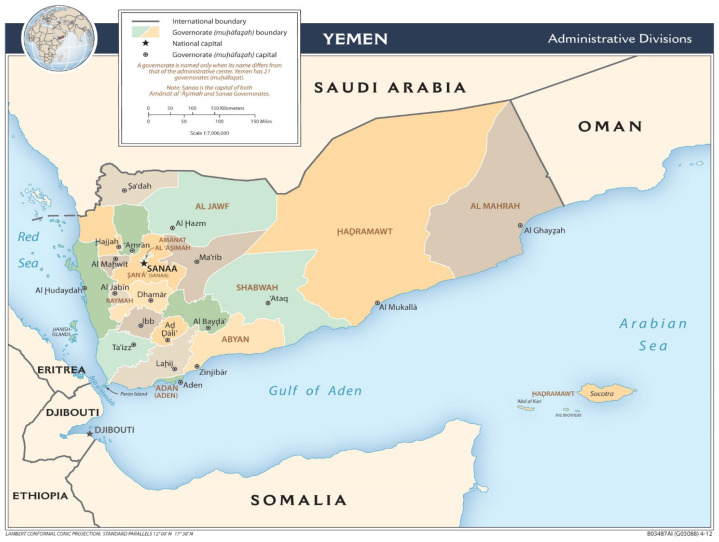
Map of Yemen. Source: Free Map Viewer. [24].

**Figure 3 vaccines-12-00311-f003:**
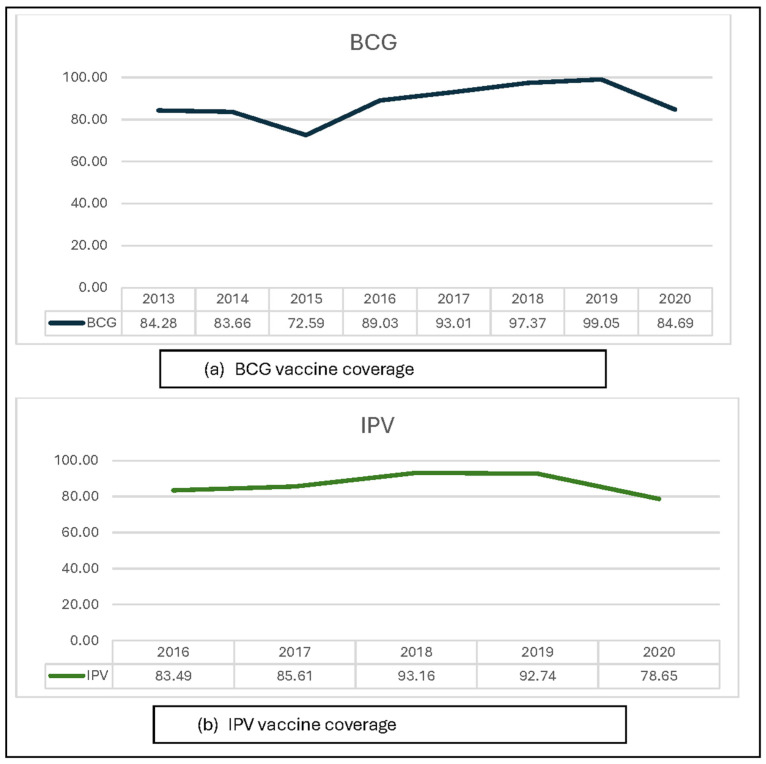
Vaccination coverage of single-dose vaccines (BCG (**a**) and IPV (**b**)) in Coastal Hadhramaut in 2013–2020.

**Figure 4 vaccines-12-00311-f004:**
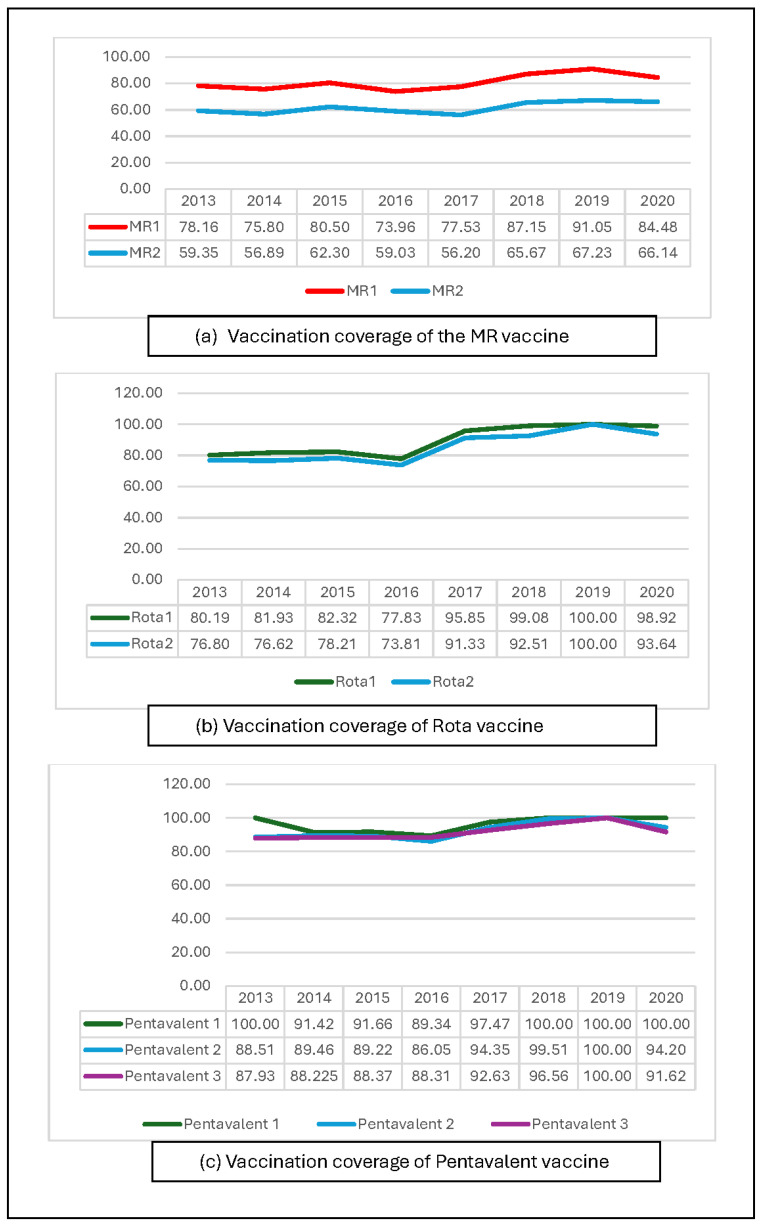
Vaccination coverage of multidose vaccines (MR (**a**), rotavirus (**b**), and pentavalent (**c**)) in Coastal Hadhramaut in 2013–2020.

**Table 1 vaccines-12-00311-t001:** Routine immunization schedule for children in Yemen.

Age of the Child	Vaccine	Remarks
After birth	BCG (single dose), OPV	BCG (essential dose) OPV (non-essential dose)
6 weeks	OPV dose 1, pentavalent vaccine, PCV, rotavirus vaccine	Administer BCG if not administered previously
10 weeks	OPV dose 2, pentavalent vaccine, PCV, rotavirus vaccine	The minimum interval between doses 1 and 2 is 4 weeks
14 weeks	OPV dose 3, pentavalent vaccine, PCV, IPV	The minimum interval between doses 2 and 3 is 4 weeks
9 months	MR dose 1 + vitamin A (100 UI) + OPV (non-essential dose)	The minimum age is 9 months
18 months	MR dose 2 + vitamin A (200 UI) + OPV (non-essential dose)	The minimum age is 18 months

## Data Availability

All data relevant to this manuscript are included in the manuscript text.

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
