# Peer review of "Immunization of Children under 2 Years Old in the Coastal Hadhramaut Governorate, Yemen, during Public Health Emergencies: A Trend Analysis of 2013–2020"

_vaccines, 2024, doi:10.3390/vaccines12030311_

Round 1
Reviewer 1 Report
Comments and Suggestions for Authors
The authors have conducted a data analysis of the EPi over a period, interspersed with several factors that posed a problem to the program.
They have addressed several contributing factors and have tried to analyze the cause.
However, as the authors have pointed out, a major limitation of the study is the absence of a denominator to apply any statistical tool for a mathematical conclusion.
An important point for poor vaccination, which needs to be looked into during natural or manmade calamities, is the break in the supply chain cycle and the availability of trained manpower.
Some other comments have been marked in the manuscript.

Author Response
please see the attached file 'response to Reviewer 1'

Reviewer 2 Report
Comments and Suggestions for Authors
I read the manuscript entitled "Childhood Immunization Status in Hadhramout Governorate, Yemen, during Public Health Emergencies: A Trend Analysis of 2013–2020" with interest. However, I found myself somewhat disappointed as I did not seem to find anything [substantially] new in the present analysis in comparison to what is already known about this topic. For example, the authors said, "Other studies done in Yemen also supported our results" in line 245. Let me clarify my point about what I mean "substantially" new.
First off, the authors had vaccination coverage data for 12 districts as they said in line 128. However, all the time-series plots they prepared and have displayed in Figures 3 and 4 are for the entire coastal Hadhramout Governorate (i.e., there is one point per year). Why? This line of investigation does not allow making any inference about the level of heterogeneities in the coverage trends, which might be present among the 12 districts and that would have been unraveled and the reason(s) behind the level of heterogeneities would have discussed or hypothesized. Or do the authors think that all 12 districts are same in every socio-demographic aspect? I therefore see it as a missed opportunity for making a good and new contribution to the literature on trends in childhood vaccination coverages in low-and-middle-income countries. To me, in fact, it seems more than a missed opportunity, given the authors had planned to analyze the childhood vaccination coverage data since February 2021 (see the date mentioned in line 368). Drawing the single-point-per-year time-series plots that are shown and discussed in the manuscript, I guess, should not have taken more than 2 years (note that I received the review invitation late in January of 2024).
The other comments are as follows:
The abstract is written well: it is somewhat focused and can convey the core message to potential readers. However, the introduction section is too long. It takes more than 3 paragraphs to come to the subject matter of this manuscript. The authors should have highlighted [and focused their thoughts on] what is known (or not known) about the trend in childhood vaccination coverages in Yemen. They started saying this in the fourth paragraph that started in line 68. Even in that paragraph the authors do not seem to be interested in mentioning anything known about the trend in childhood vaccination coverages in Yemen. I strongly suggest that authors should shorten the introduction section by providing only the necessary background information and coming to the focus of their analysis early on (in the first paragraph itself).
Why the y-axis labels in some plots (e.g., Figure 3a) go beyond 100%? Vaccine coverage cannot be more than 100%. The authors need to change the y-axis label in these plots. Also, make sure that the y-axis range starts from the same minimum (e.g., 0) to 100% in all plots of Figures 3 and 4.
When I read the points 4.1 through 4.4 in the Discussion section, I realized that the authors had relied too much on previous works carried out by others. The authors have not done any analysis regarding these specific points. For example, in 4.1, the authors seem to make a point that parental vaccine hesitancy might have contributed to low (or decrease in) vaccine coverage in Yemen’s Coastal Hadhramout Governorate. They end the paragraph with a sentence: “Other studies done in Yemen also supported our results [12,29].” By ending the paragraph with this sentence, I am not sure which results [from which years] the authors are referring to. Their analysis of the vaccine coverage data could not show any signature of parental vaccine hesitancy because the data do not have anything on the parental vaccine hesitancy. In contrast, if one looks at the time series of the vaccine coverage (shown in Figures 3 and 4), then for many vaccines, they could see that the coverages have steadily increased until 2020. Will not this then mean that parents were cooperative (and hence less hesitant) in getting their children vaccinated? I therefore encourage the authors to be careful while discussing something which they themselves have not analyzed in this paper. This is important because this work is an original research (not a review paper). Any discussion around the results presented in this paper should be related to the findings that the authors have. I hope the authors should apply this suggestion to each of these discussion points during the revision.
The title of the manuscript does not seem to be accurate. The trend analysis has been done for the coastal districts of Hadhramout Governorate, and not for the entire Hadhramout Governorate. The title therefore should reflect that fact.
In Table 1, the age of children is shown to range from their birth up to 18 months. Does this mean that the time-series of vaccine coverages are for children up to 18 months old? If this is the case, then this fact should be made clear in the title and the Material and Methods sections and in the Discussion section the results should be appropriately compared with other published results.
The authors wrote “Our deepest condolences to the Director of the Immunization Program, Fouad Ali Bamatref, who recently passed away.” This statement should not be in the acknowledgement section. Because this is not an acknowledgement. If the authors were helped by the Director, then that help should be acknowledged.
Comments on the Quality of English LanguageThe revised manuscript should be thoroughly proofread.
Author Response
please see the attachment file's response to the reviewer 2'

Reviewer 3 Report
Comments and Suggestions for Authors
In this study the authors report on the impact of political unrest, humanitarian crises, and the COVID-19 pandemic on childhood vaccination rates in Yemen. After evaluating vaccination data collected from the Coastal Hadhramaut Governorate in Yemen from 2013-2020 the authors report that vaccination rates for BCG, MR, and pentavalent declined following the start of the civil war as well as the COVID-19 pandemic. While this study is predominantly descriptive, it importantly highlights concerns with falling vaccination rates in child in Yemen. Further, the authors provide several compelling explanations for these declines which in turn inform potential approaches to combat further declines in childhood vaccination rates.
Suggestions:
1. The graphs clearly show trends in coverage. However, it would be more informative and impactful to show data into a table form- both vaccination rate and % change (+/-) from the previous year.
2. The manuscript would be more impactful if data on the cases/rates of the relevant diseases was presented (if available).
Author Response
please see the attachment file 'response to reviewer 3'

Round 2
Reviewer 2 Report
Comments and Suggestions for Authors
I thank the authors for submitting the revised manuscript. I am happy to see that the authors have revised their manuscript following the suggestions of the reviewers. However, I am still concerned about the following:
In the previous review comments, I wrote: "I strongly suggest that authors should shorten the introduction section by providing only the necessary background information and coming to the focus of their analysis early on (in the first paragraph itself)." The revised introduction section is almost left unchanged. Still, it is too long. Some of the statements or paragraphs do not seem to be directly relevant to (or in the context of) the analysis presented in the manuscript. For example, general information provided in the first paragraph does not seem to be necessary for potential users/readers of the results, analyses and their interpretation provided in the manuscript. Likewise, a statement like this,
"A public health emergency is defined as ... “, in lines 71 - 81, is not at all needed to understand the results and the main focus of the paper. The paper is not [and it should not be] about educating readers what public health emergency is or how it is defined. The authors therefore must shorten the introduction section by [focusing] on what the results, analyses and their interpretation presented in the manuscript are likely to add to the knowledge gaps that might exist currently in the trends of "immunization of Children Under 2 Years Old in the Coastal Hadhramout Governorate, Yemen." The authors should ask themselves: have other researchers presented similar results and analyses for Yemeni children, especially under 2 years old?
Also, the authors have not corrected the vaccination coverages being above 100%. They have provided some explanation as to why this would be the case. But the explanation would not allow them to say that the coverage would be higher than the 100% mark, as per my viewpoint. I suggest whenever the coverage goes above the 100% mark, it should be capped at 100%. And the authors should mention this correction in the methods section. Or else the authors must find the correct denominators (i.e., children's population) for those years.
Author Response
Comment 1: In the previous review comments, I wrote: "I strongly suggest that authors should shorten the introduction section by providing only the necessary background information and coming to the focus of their analysis early on (in the first paragraph itself)." The revised introduction section is almost left unchanged. Still, it is too long. Some of the statements or paragraphs do not seem to be directly relevant to (or in the context of) the analysis presented in the manuscript. For example, general information provided in the first paragraph does not seem to be necessary for potential users/readers of the results, analyses and their interpretation provided in the manuscript.
Likewise, a statement like this, "A public health emergency is defined as ... “, in lines 71 - 81, is not at all needed to understand the results and the main focus of the paper. The paper is not [and it should not be] about educating readers what public health emergency is or how it is defined. The authors therefore must shorten the introduction section by [focusing] on what the results, analyses and their interpretation presented in the manuscript are likely to add to the knowledge gaps that might exist currently in the trends of "immunization of Children Under 2 Years Old in the Coastal Hadhramout Governorate, Yemen." The authors should ask themselves: have other researchers presented similar results and analyses for Yemeni children, especially under 2 years old?
Response 1: Thanks for the comment. The following change has been made in the introduction:
1- we deleted some unnecessary information in the first paragraph (line 42-45, 46-47)
2- we deleted unnecessary information in the fourth and fifth paragraph (line 73-76, 87-89,100-101,113-114)
Comment 2: Also, the authors have not corrected the vaccination coverages being above 100%. They have provided some explanation as to why this would be the case.
But the explanation would not allow them to say that the coverage would be higher than the 100% mark, as per my viewpoint. I suggest whenever the coverage goes above the 100% mark, it should be capped at 100%. And the authors should mention this correction in the methods section. Or else the authors must find the correct denominators (i.e., children's population) for those years.
Response 2: Thanks for the comment, we considered your suggestion and we capped the coverage to be 100%. Also, we mentioned this correction in the method section in lines: 186 -195).